# Shrinkage-based Bias-Variance Trade-off for Deep Reinforcement Learning

## Abstract

Deep reinforcement learning has achieved remarkable successes in solving various challenging artificial intelligence tasks. A variety of different algorithms have been introduced and improved towards human-level performance. Although technical advances have been developed for each individual algorithms, there has been strong evidence showing that further substantial improvements can be achieved by properly combining multiple approaches with difference biases and variances. In this work, we propose to use the *James-Stein* (JS) shrinkage estimator to combine on-policy policy gradient estimators which have low bias but high variance, with low-variance high-bias gradient estimates such as those constructed based on model-based methods or temporally smoothed averaging of historical gradients. Empirical results show that our simple shrinkage approach is very effective in practice and substantially improve the sample efficiency of the state-of-the-art on-policy methods on various continuous control tasks.

## 1 Introduction

Deep reinforcement learning (RL) has achieved remarkable successes recently, as witnessed by the human/super-human level performance achieved in various challenging artificial intelligence tasks; notable examples include AlphaGo (Silver et al., 2017; 2016), Atari games (Mnih et al., 2013), robotic learning (Levine et al., 2016; Andrychowicz et al., 2017), among many others. A remarkable nature of RL is that multiple types of algorithms have been developed, based on different principles, highlighting different features and trade-offs. This includes model-based approaches which derive optimal policies from estimated dynamic models of the unknown environment (Deisenroth & Rasmussen, 2011; Nagabandi et al., 2017), model-free off-policy methods such as Q-learning (Mnih et al., 2013), as well as model-free on policy methods such as Proximal Policy Optimization (PPO) (Schulman et al., 2017) and Trust Region Policy Optimization (TRPO) (Schulman et al., 2015). These methods exploit different properties of the Markov decision process, and have different pros and cons. Model-based and off-policy methods are known to be more sample efficient when implemented successfully, while on-policy methods are less sample efficient, but are simpler and easier to train, especially for problems with continuous or high dimensional action spaces. Although there have been emerging advances in improving each individual algorithms, more substantial improvements are likely achieved by adaptively integrating multiple strategies into an overall intelligence system. This is evident from human brain, which has been discovered to contain multiple distinct or even competing pathways for learning from reward (Daw et al., 2005; Rangel et al., 2008), including both model-free strategies consisting of error-and-trial learning to repeat rewarded actions, and model-based learning that learns models of the environment, and use it to evaluate candidate actions by mental simulations of their consequences. We would expect integrating different methods in RL will improve the performance, especially when integrated into an adaptive fashion. Algorithmically, a key challenge for an adaptive combination is to evaluate the reliability of each individual strategy, without knowing the ground truth answer while performing the task.

In this work, we focus on the problem of adaptively combining policy gradient estimators from various sources. We leverage the fact that different strategies yield different gradient estimators with different bias-variance trade-off, which can be exploited by statistical shrinkage estimators to obtain optimal combinations that have better mean square error than any individual estimators. In particular, policy gradient estimators provide (nearly) unbiased estimators of the true gradient of the expected reward, but often suffers from high variance and is sample inefficient. Model-based and on-policy

methods, on the other hand, provide more biased estimates but can have much smaller variance by taking the advantage of the large amount of imaginary data efficiently simulated from the model, or off-policy data stored in the replay buffer.

In particular, we propose to use *James-Stein* (JS) shrinkage estimator (Stein, 1956; James & Stein, 1961; Efron & Morris, 1975) to combine the unbiased on-policy gradient estimator with biased but low variance estimators constructed based on models or off-policy data. The basic idea of JS estimator is to access the quality of the biased estimator by inspecting its consistency with unbiased estimators, and use it as a clue for constructing an optimally weighted combination to trade-off bias and variance. JS estimator admits strong theoretical guarantees for standard Gaussian cases, and more importantly, also provides a simple practical heuristic for more complex practical non-Gaussian cases. It can potentially provide an ideal tool for solving the very challenge of bias-variance trade-off in various components of deep reinforcement learning, but seems to be still largely under-explored in the literature. In this work, we explore the practical application of JS estimators in deep RL, illustrated by two examples:

**Shrinking Towards Model-based Gradient**   We develop a hybrid algorithm which adaptively integrates on-policy policy gradient with Dyna-style model-based learning agent (Sutton, 1991). We train a transition model using the existing rollout data in parallel to the on-policy gradient descent; imaginary rollouts are simulated from the model, providing a biased, lower variance gradient estimator. This model-based gradient estimator compared against the unbiased, high variance on-policy gradient, and a weighted combination is constructed using JS estimator based on the consistency of the two estimators. Our method improves the sample efficiency by exploiting the data more efficiently using model learning while avoiding significant bias introduced by the model using adaptive JS estimator.

**Shrinking Towards Temporally Smoothed Gradient**   It has been observed heuristically that proper bias is helpful for training (Sutskever et al., 2013; Nesterov, 2013). We present a simple method that explicitly constructs an optimal bias-variance trade-off by shrinking the on-policy gradient estimator towards a smoothed gradient estimator obtained by a weighted sum over previous iterations. The smoothed gradient provides a reasonable guess of the true gradient of the current estimation and forms a biased, low variance estimator. By shrinking towards the smoothed gradient, we significantly decrease the variance of the on-policy gradient, at the cost of introducing a small bias. This effectively creates a *statistical momentum*, which shares similarity with the traditional momentum methods derived from optimization perspectives, but with an adaptive coefficient determined by the JS estimator. It can also be combined with traditional momentum to yield gains on both statistical and optimization sides. This simple algorithm effectively leverages the historic (off-policy) data via the temporally smoothed gradient, yielding higher sample efficiency than standard policy gradient.

We perform empirical studies to test the performance of both above examples, showing our adaptive combination strategies can yield better results than any fixed-weight combination, significantly improving the sample efficiency on standard on-policy methods. This illustrates the practical power of JS estimators, which we expect to be broadly applicable to more settings in deep RL in future works.

## 2   BACKGROUND

We first introduce the basics of reinforcement learning and model-free policy gradient, and set up the notation that we will use in the paper. We then introduce a model-based policy gradient method.

### 2.1   REINFORCEMENT LEARNING

Reinforcement learning considers the problem of finding an optimal policy for an agent which interacts with an uncertain environment and collects reward per action. The goal of the agent is to maximize the long-term cumulative reward. Formally, this problem can be formulated as a Markov decision process over the environment states $s \in S$ and agent actions $a \in A$, under an unknown environmental dynamic defined by a transition probability $T(s'|s, a)$ and a reward signal $r(s, a)$ immediately following the action $a$ performed at state $s$. The agent's action $a$ is selected by a conditional probability distribution $\pi(a|s)$ called policy. In policy optimization, we consider a set of candidate policies $\pi_\theta(a|s)$ parameterized by $\theta$ and obtain the optimal policy by maximizing the

expected cumulative reward or return

$$J(\theta) = \mathbb{E}_{s \sim \rho_\pi, a \sim \pi(a|s)} \left[ r(s,a) \right],$$

where $\rho_\pi(s) = \sum_{t=1}^{\infty} \gamma^{t-1} \Pr(s_t = s)$ is the normalized discounted state visitation distribution with discount factor $\gamma \in [0,1)$. To simplify the notation, we denote $\mathbb{E}_{s \sim \rho_\pi, a \sim \pi(a|s)}[\cdot]$ by simply $\mathbb{E}_\pi[\cdot]$ in the rest of paper.

## 2.2 ON-POLICY POLICY GRADIENT

According to the policy gradient theorem (Sutton & Barto, 1998), the gradient of $J(\theta)$ equals

$$\nabla_\theta J(\theta) = \mathbb{E}_\pi \left[ \nabla_\theta \log \pi(a|s) Q^\pi(s,a) \right], \tag{1}$$

where $Q^\pi(s,a) = \mathbb{E}_\pi \left[ \sum_{t=1}^{\infty} \gamma^{t-1} r(s_t, a_t) | s_1 = s, a_1 = a \right]$ denotes the expected return under policy $\pi$ starting from state $s$ and action $a$.

Different policy gradient methods are based on different stochastic estimations of the expected gradient in Eq (1). Perhaps the most straightforward way is to simulate the environment with the current policy $\pi$ to obtain a trajectory $\{(s_t, a_t, r_t)\}_{t=1}^{n}$ and estimate $\nabla_\theta J(\theta)$ using the Monte Carlo estimation:

$$\hat{\nabla}_\theta J(\theta) = \frac{1}{n} \sum_{t=1}^{n} \gamma^{t-1} \nabla_\theta \log \pi(a_t|s_t) \hat{Q}^\pi(s_t, a_t), \tag{2}$$

where $\hat{Q}^\pi(s_t, a_t)$ is an empirical estimate of $Q^\pi(s_t, a_t)$, e.g., $\hat{Q}^\pi(s_t, a_t) = \sum_{j \geq t} \gamma^{j-t} r_j$. This is known as an *on-policy* gradient estimator because it only uses the data from the current policy.

Proximal Policy Optimization (PPO) (Schulman et al., 2017; Heess et al., 2017) is one of the state-the-art model-free policy gradient methods for policy optimization. It uses a proximal Kullback-Leibler (KL) divergence penalty to regularize and stabilize the policy gradient update. Given an existing policy $\pi_{\text{old}}$, PPO obtains a new policy by maximizing the following surrogate loss function

$$J_{\text{ppo}}(\theta) = \mathbb{E}_{\pi_{\text{old}}} \left[ \frac{\pi_\theta(a|s)}{\pi_{\text{old}}(a|s)} Q^\pi(s,a) - \lambda \text{KL} \left[ \pi_{\text{old}}(\cdot|s) \;\|\; \pi_\theta(\cdot|s) \right] \right],$$

where the first term is an approximation of the expected reward, and the second term enforces the the updated policy to be close to the previous policy under KL divergence. The gradient of $J_{\text{ppo}}(\theta)$ can be rewritten as

$$\nabla_\theta J_{\text{ppo}}(\theta) = \mathbb{E}_{\pi_{\text{old}}} \left[ w_\pi(s,a) \nabla_\theta \log \pi(a|s) Q_\lambda^\pi(s,a) \right]$$

where $w_\pi(s,a) := \pi_\theta(a|s)/\pi_{\text{old}}(a|s)$ is the density ratio of the two polices, and $Q_\lambda^\pi(s,a) := Q^\pi(s,a) + \lambda w_\pi(s,a)^{-1}$ where the second term comes from the KL penalty.

The advantage of using on-policy estimation is that it provides (nearly) unbiased estimation of the gradient of expected reward. A well-known challenge, however, is that it has large variance, given that it is only possible to use a small number of data from the current policy at each iteration because the cost of simulation from the real environment is high. In addition, it does not take advantage of the off-policy data collected previously. Using off-policy data allows us to obtain low variance but potentially highly biased estimates, which can be combined with on-policy gradient with James-Stein estimator as we propose in this work. In the sequel, we introduce model-based policy gradient as an important way for leveraging off-policy data.

## 2.3 MODEL-BASED POLICY GRADIENT

The model-based methods derive optimal polices by learning a transition model of the underlying dynamic $T(s'|s,a)$ of the Markov Decision Process. Assume the states $s$ is continuous, it is common to assume $s' = s + f_\phi(s,a) + \xi$ where $f_\phi$ is a parametric function (e.g., neural network) with learnable parameter $\phi$, and $\xi$ is a zero-mean noise. The model parameter $\phi$ is often trained by minimizing the $L_2$ one-step prediction loss,

$$\min_\phi \sum_{t=1}^{n} \|s_{t+1} - s_t - f_\phi(s_t, a_t)\|_2^2, \tag{3}$$

where $D := (s_t, a_t, s_{t+1})_{t=1}^n$ is a set of transition pairs collected from previous rollouts. See Deisenroth & Rasmussen (2011); Fu et al. (2016); Nagabandi et al. (2017); Thanard Kurutach (2018); Depeweg et al. (2017) for variety of the recent variants of model-based learning methods.

With a learned model, the problem of estimating the optimal policy becomes a planning problem, which can be solved either using various classical planning and control methods, typically based on variants of dynamic programming (see e.g., Sutton & Barto, 1998), or sample-based methods that generate fictitious data by simulating the model forward and obtain optimal policies by applying standard RL algorithm on the fictitious data.

In this work, we will mainly consider a simple sample-based policy gradient method whose main idea is rooted from Dyna (Sutton, 1991). The idea is simply generating a fictitious data $(\tilde{s}_t, \tilde{a}_t, \tilde{r}_t, \tilde{s}_{t+1})$ by simulating the model, and plug it into the policy gradient formula (2),

$$\hat{\nabla}_\theta^{\mathrm{model}} J(\theta) = \sum_{t=1}^n \gamma^{t-1} \nabla_\theta \log \pi(\tilde{a}_t | \tilde{s}_t) \hat{Q}^\pi (\tilde{s}_t, \tilde{a}_t). \tag{4}$$

By simulating a large number of fictitious data from the model, which is much less expensive than simulating from the real environment, we can make the variance of $\hat{\nabla}_\theta^{\mathrm{model}} J(\theta)$ significantly smaller than the variance of the on-policy gradient in (2).

The idea of using $\hat{\nabla}_\theta^{\mathrm{model}} J(\theta)$ can be found in (Gu et al., 2016; Thanard Kurutach, 2018). The problem, however, is that the transition model can deviate significantly from the true model, introducing a potentially large bias that makes the training problem unstable. Thanard Kurutach (2018) found that by training an ensemble of models can help stabilize the training. As we discuss in the sequel, one of our key idea is to calculate on-policy and model-based gradient simultaneously, and use the unbiased information of the on-policy gradient to evaluate the biasness of model-based gradient, and weigh its importance accordingly to ensure the stability while exploiting the model-based information.

## 3 Gradient Estimation with Shrinkage

We introduce our main method which uses shrinkage estimator to trade-off the bias and variance to adaptively combine multiple gradient estimators. We start with reviewing James-Stein Shrinkage estimator in Section 3.1, and then discuss its two applications in reinforcement learning, including shrinkage estimator towards model-based policy gradient in Section 3.2, and temporally smoothed gradients in Section 3.3.

### 3.1 James-Stein Shrinkage Estimator

James-Stein estimator (Stein, 1956; James & Stein, 1961) is an estimator of the mean of a multivariate normal distribution that achieves a smaller mean square error (MSE) than the maximum likelihood estimator (MLE) in dimensions more than three. This estimator shrinks the standard maximum likelihood estimator towards a specified target by an adaptive shrinkage factor, which introduces bias whilst obtains lower variance and MSE.

Consider the problem of estimating the mean of a $p$-dimensional multivariate normal distribution $\mathcal{N}(\theta, Q)$, where $\theta \in \mathbb{R}^d$ is the mean and $Q \in \mathbb{R}^{p \times p}$ is the covariance matrix. Given a sample $\boldsymbol{x}$ drawn from $\mathcal{N}(\theta, Q)$, a natural estimator of the maximum likelihood estimation (MLE) which predicts $\theta$ using $\theta \approx \boldsymbol{x}$. This estimator is unbiased, and gives the minimum variance among all the possible unbiased estimators.

MLE was used to be believed to be admissible, that is, no other estimators can always achieve lower MSE than MLE. However, this was proved to be false by James and Stein, who constructed an estimator, now known as James-Stein estimator, that always dominants MLE when $p > 2$:

$$\hat{\theta}^{\mathrm{JS}} = \bar{\theta} + \alpha(\boldsymbol{x} - \bar{\theta}), \tag{5}$$

where $\bar{\theta} \in \mathbb{R}^p$ is any fixed constant relative to $\boldsymbol{x}$, and $\alpha$ is a combination coefficient defined as

$$\alpha = 1 - (\tilde{p} - 2)/(\boldsymbol{x} - \bar{\theta})^\top Q^{-1} (\boldsymbol{x} - \bar{\theta}), \tag{6}$$

---

**Algorithm 1** PPO-MBS: PPO with model-based gradient shrinkage

---

Initialize policy $\pi_\theta(a|s)$, model-based dynamic $f_\phi(s,a)$, and replay buffer $\mathcal{D} \leftarrow \emptyset$.
**repeat**
    Collect real trajectories using $\pi_\theta$ and add them to replay buffer $\mathcal{D}$.
    **for** K iterations **do**
        Update $\phi$ with data from replay buffer $\mathcal{D}$ using (3).
    **end for**
    Sample imaginary trajectories $\{\tau_i\}_{i=1}^n$ with max time-step $T_p$ using $\pi_\theta$ and $f_\phi$.
    **for** M iterations **do**
        Calculate model free PPO gradient $g_c$ w.r.t $\theta$ with real samples.
        Calculate model-based PPO gradient $g_{mb}$ w.r.t $\theta$ with imaginary trajectories $\{\tau_i\}_{i=1}^n$.
        Get JSE $\hat{g}^{JS^+}$ using (8) and update $\theta$ with $\hat{g}^{JS^+}$.
    **end for**
**until** Convergence

---

where $\tilde{p} = \text{trace}(Q)/\lambda_{\max}(Q)$ is considered as an effective dimension of covariance matrix $Q$, which equals $p$ if $Q$ is an identity matrix. The intuition is that $\ell := (\boldsymbol{x} - \bar{\theta})^\top Q^{-1}(\boldsymbol{x} - \bar{\theta})$ the bias of $\bar{\theta}$ in respect to the variation of $\boldsymbol{x}$. If $\bar{\theta}$ is close to the true mean $\theta$, $\alpha$ is small and $\bar{\theta}$ is highly weighted; if $\bar{\theta}$ is far away from the true mean, $\alpha$ is large and the unbiased estimator $\boldsymbol{x}$ is highly weighted. Overall, the adaptive strategy ensures the bias and variance is always traded off optimally.

More generally, we can relax $c = \tilde{p} - 2$ in equation (5) to $0 < c < 2(\tilde{p} - 2)$, which still dominates the MLE when $\tilde{p} > 2$. Baranchik (1964) also suggests that JS estimator can be further improved by limiting the combination coefficient to be positive. This gives a general positive James-Stein estimator: (JSE$^+$) as

$$\hat{\theta}^{JS^+} = \bar{\theta} + \alpha^+(\boldsymbol{x} - \hat{\theta}), \qquad \alpha^+ = \max\left(0,\ 1 - c/(\boldsymbol{x} - \bar{\theta})^\top Q^{-1}(\boldsymbol{x} - \bar{\theta})\right). \qquad (7)$$

Although the standard theoretical results of JS estimators are established for normal distribution, which may not hold for settings in deep RL, however, JS still provides a simple yet highly useful practical strategy for adaptive combination. In practice, when the covariance matrix $Q$ is not known, it can be replaced by an empirical estimation $\hat{Q}$, which is estimated over multiple trajectories. In fact, in our subsequent experiments, we approximate $\hat{Q}$ with its diagonal matrix to avoid matrix inverse. This introduces error in estimating $\alpha^+$, but can be practically compensated choosing a proper parameter $c$ using trial tests. This work focuses on demonstrating a novel empirical application of JS estimators for deep RL, and defers further theoretical work to future works.

### 3.2 Shrinking Towards Model-based Gradient Estimator

We describe our method that adaptively combines model-based and on-policy strategies using shrinkage estimator. For notation, we denote by $g_c := \hat{\nabla}_\theta J(\theta)$ the on-policy gradient in (2) obtained from the "real" trajectories, and $g_{mb} := \hat{\nabla}_\theta^{model} J(\theta)$ the model-based gradient obtained from "imaginary" trajectories rolled out from learned dynamics. With these two gradient estimators, we can obtain a new estimator as

$$\hat{g}^{JS^+} = g_{mb} + \alpha^+(g_c - g_{mb}). \qquad (8)$$

The only difference between these two estimators $g_{mb}$ and $g_c$ is the data used for gradient calculation. Here $\alpha^+$ serves as a measurement of the quality of dynamic model training. A good dynamic gives imaginary trajectories that are similar to real trajectories, making $g_{mb}$ are close to $g_c$ and hence a small $\alpha^+$ according to (6).

One subtle difficulty is that when simulating long horizon trajectories, the error in the model is amplified, causing significantly bad model-based estimator (yielding $\alpha \approx 1$ in (8)). To address this problem, we simulate only short horizon imaginary trajectories initialized with different states randomly drawn from true samples. This trick allows us to obtain higher quality model-based estimators. We integrate our method with Proximal Policy Optimization (PPO)(Schulman et al., 2017; Heess et al., 2017), which we refer as `PPO-MBS`, and is summarized in algorithm 1.

---

**Algorithm 2** PPO-STS: PPO with temporally smoothed gradient shrinkage

---

Initialize policy $\pi_\theta(a|s)$ and historic gradient set $\beta \leftarrow \emptyset$.
**repeat**
    Collect real trajectories using $\pi_\theta$.
    **for** M iterations **do**
        Calculate model free PPO gradient $g_c$ w.r.t $\theta$.
        Calculate weighted average historic gradient $g_w$.
        Get JSE $\hat{g}^{JS^+}$ using (10) and update $\theta$ with $\hat{g}^{JS^+}$.
        Add $g_w$ to the gradient set $\beta$.
    **end for**
**until** Convergence

---

### 3.3 SHRINKING TOWARDS TEMPORALLY SMOOTHED GRADIENT

In addition to the model-based method, we present another simpler method that constructs shrinkage estimator based on a smoothed gradient obtained by averaging previously gradients. This simple strategy is significantly simpler and computationally efficient since no additional model learning or other expensive calculation is needed. It turns out this strategy works surprisingly well.

Denote by $g_t$ the on-policy gradient at iteration $t$, and $g_c := g_T$ the gradient at the current iteration ($T$). We calculate a smoothed gradient from the previous iterations by a weighted average:

$$g_w = \left( \sum_{t<T} \rho^t g_t \right) \bigg/ \left( \sum_{t<T} \rho^t \right). \tag{9}$$

Assume the parameters do not change much over the iterations, $g_w$ provides a reasonable guess of the current gradient. Therefore, we can shrink the current noisy gradient towards the $g_w$:

$$\hat{g}^{JS^+} = g_w + \alpha^+(g_c - g_w). \tag{10}$$

We refer our method as `PPO-STS` and summarize it in algorithm 2. Interestingly, one can view the update in (10) as a type of momentum method, because it forces the parameters to update along the previous gradient directions. Compared with the standard momentum methods developed from the optimization perspective (referred as *optimization momentum*) (Sutskever et al., 2013; Nesterov, 2013; Kingma & Ba, 2014; Duchi et al., 2011), our method is derived for the statistical purpose of obtaining a better estimator of gradient (and hence forms *statistical momentum*). A unique feature of the statistical momentum is that its magnitude changes adaptively with the consistency between the momentum and the current gradient estimator, reducing to zero if the gradient estimator is deterministic (zero variance). As we do in our experiments, our statistical momentum can be directly applied on the top of classical optimization momentum to combine the benefits of both.

## 4 EXPERIMENTS

We evaluate PPO-MBS and PPO-STS on continuous control tasks (Brockman et al., 2016) using the MuJoCo physics simulator (Todorov et al., 2012). We report the variance, bias, and MSE of gradient estimation with and without PPO-STS, PPO-MBS in Section 4.1, and then we show the results of extensive experiments in Section 4.2. We choose Proximal Policy Gradient (PPO) as the baseline, which is one of the state-of-the-art policy gradient methods on various continuous control tasks. Implementation details[1]and hyperparameters values are provided in Appendix A and Table 1.

Extensive experiments show that PPO-STS and PPO-MBS can reduce the MSE and the variance of the gradient estimation while we can obtain significantly better performance comparing standard on-policy methods.

### 4.1 EVALUATION OF PPO-STS AND PPO-MBS

We start with evaluating the effect of PPO-STS and PPO-MBS on gradient estimation, namely, we demonstrate the bias variance decomposition of gradient estimations, and also compare the average reward of adaptive combination with different fixed combinations.

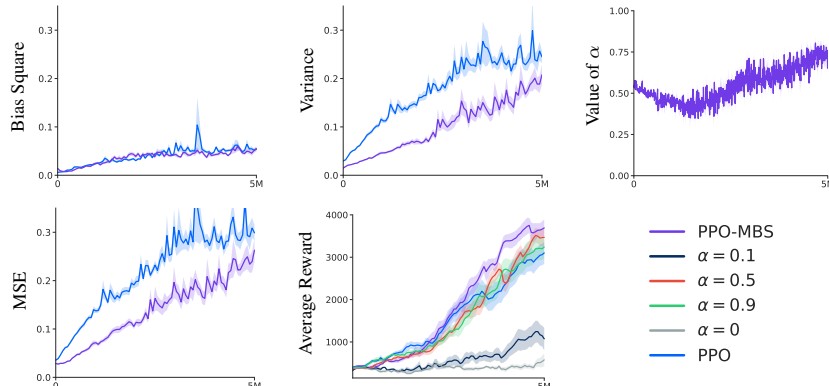

Figure 1: Comparison of MSE, Variance, and Bias of PPO and PPO-MBS gradient estimation on Walker2d.

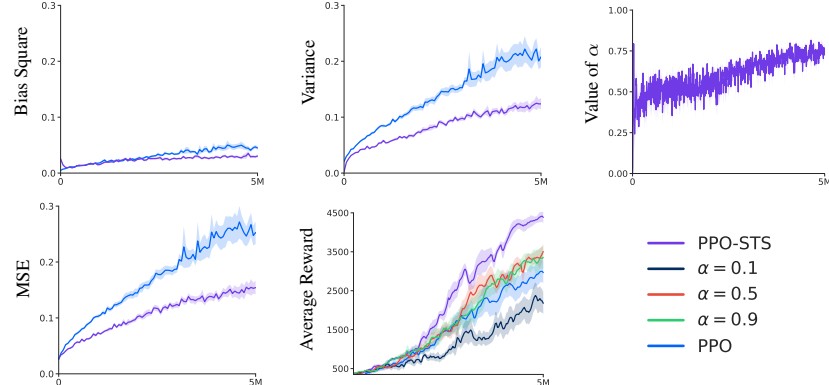

Figure 2: Comparison of MSE, Variance, and Bias of PPO and PPO-STS gradient estimation on Walker2d.

We choose the Walker2d as the evaluation environment, and calculate the MSE, variance, and bias of gradient estimation in PPO-MBS (Algo 1), PPO-STS (Algo 2), and PPO during the training process. Figure 1 shows the results on Walker2d. Similarly, Figure 2 shows the results of PPO-MBS on Walker2d-v1. The empirical results show that our methods obtain significantly lower MSE, and variance than the typical model-free policy gradient method. It also shows that adaptive combination (adaptively to learn value of $\alpha$) is essential to get a better estimation of gradient, and can lead to higher reward in policy optimization. The value of James-Stein coefficient $\alpha$ can also indicate the difference between temporal gradient and current gradient, or model-free policy gradient and model-based policy gradient estimates.

### 4.2 COMPARISON PPO-STS, PPO-MBS WITH STATE-OF-THE-ART METHODS

Finally, we evaluate PPO-STS, PPO-MBS on a more extensive list of tasks shown in Figure 3. It shows that PPO-STS and PPO-MBS have superior performance over baseline method PPO. Empirically, we find that PPO-STS has better performance on high dimensional tasks, such as Walker2d while PPO-MBS performs better on low dimensional environments such as Reacher, Swimmer or Hopper. Ww further investigate PPO-STS on more higher dimensional tasks such as Humanoid and Ant, the results of which are in Appendix B.

## 5 RELATED WORK

As a classical method, James-Stein (JS) estimator has been extensively studied and used in statistics and machine learning. See e.g., Gruber (2017); Young & Smith (2005) for overviews. However, we are not aware of previous works on applying JS estimators for gradient estimation in RL.

The idea of combining multiple algorithms to achieve better results have been widely considered in reinforcement learning. For example, Wiering & Van Hasselt (2008) developed an ensemble method

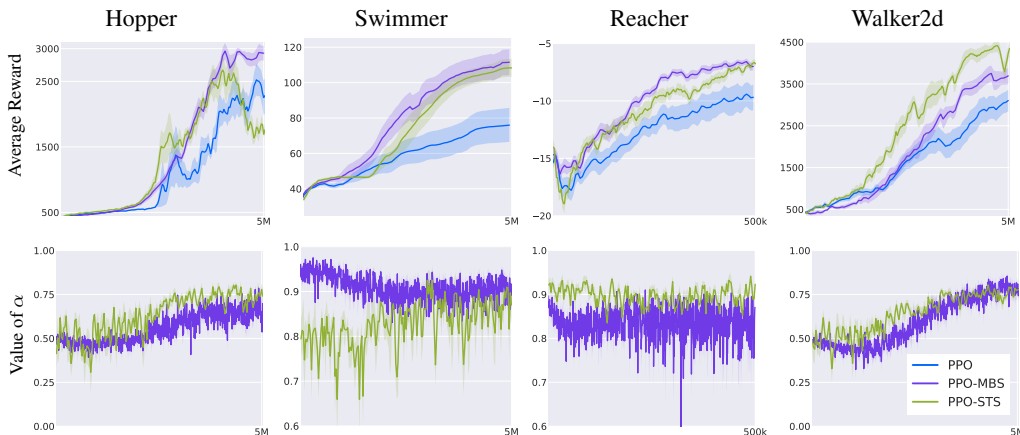

Figure 3: Comparison of PPO-STS, PPO-MBS, and PPO on various continuous control task.

by combining the policies derived from the value functions of the different RL algorithms. Hessel et al. (2017) combines several extensions of DQN to improve Atari games. Closely related to our work is Gu et al. (2017), which uses a linear combination of on-policy and off-policy gradients, but with a fixed, user-specified combination coefficient. O'Donoghue et al. (2017) unifies policy gradient and Q-Learning and empirically observes better data efficiency and stability.

Another large body of works have explored the model-based/ model-free integration. The classic Dyna framework (Sutton, 1991) has primarily been used with small and discrete systems, although its extensions for high dimensional continuous control tasks exist. (Gu et al., 2016; Feinberg et al., 2018). There is also a line of research using imaginary trajectories for training policy (Kalweit & Boedecker, 2017; Thanard Kurutach, 2018). Heess et al. (2015) uses models to improve the accuracy of model-free value function backups. Model-based training has also been used to produce a good initialization for the model-free training (Farshidian et al., 2014; Nagabandi et al., 2017), most recently, Clavera et al. (2018) proposes to meta learn a policy to adapt to an ensemble of dynamic models.

Our work provides a unified adaptive framework for various gradient estimators, with the purpose of utilizing advantages of each component, which results in lower variance and MSE over on-policy model free optimization. Two illustrated examples in our experiments show that our adaptive combination strategy achieves better results than on-policy methods or fixed-weighted combinations.

## 6 CONCLUSION

We propose to use James-Stein shrinkage estimator to combine the unbiased on-policy gradient estimator with biased but low variance estimators constructed based on models-based methods or historic temporally smoothed gradients. Empirical experiments show that our adaptive combination strategy can reduce variance and MSE, yielding significant improvement over standard on-policy methods. Future work includes investigating more powerful shrinkage estimators and alternative biased estimators for combination in policy optimization.

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

# A EXPERIMENT DETAILS

## A.1 POLICY AND VALUE FUNCTION ARCHITECTURES

The network structure largely depends on the OpenAI baselines (Dhariwal et al., 2017). The stochastic policy $\pi_\theta(a_t|s_t)$ is parameterized with a diagonal Gaussian $\mathcal{N}(\mu_\theta(s_t), \Sigma_\theta)$ with a local state-dependent mean and a global diagonal variance matrix. $\mu_\theta(s_t)$ is a two-layer MLP with hidden sizes $64 \times 64$. The value function $\hat{v}^\pi(s_t)$ is parameterized with $64 \times 64$ MLP separately and we use GAE (Schulman et al., 2016) to estimate advantage function. We use `tanh` as the activation function for both networks.

## A.2 TRAINING DETAILS

For all training methods, we use Adam (Kingma & Ba, 2014) as the gradient descent optimizer. For every 10 training iteration, we evaluate the policy $\pi_\theta(a|s)$ with its mean $\mu_\theta(s)$ using 25,000 rollout samples. The key parameters and hyperparameter search values of the training algorithms are summarized in Table 1. The optimal performing hyperparameter results are reported over 5 different seeds.

| Parameter | PPO | PPO-MBS | PPO-STS |
|---|---|---|---|
| batchsize | 5,000 (500 for *Reacher*) | | |
| learning rate of $\hat{v}^\pi(s_t)$ | 0.001 | | |
| GAE $\lambda$ and $\gamma$ | $\lambda = 0.95, \ \gamma = 0.99$ | | |
| learning rate of $\pi_\theta(a|s)$ | {1e-3, 3e-4, 1e-4} | 0.0003 | |
| inner iteration $M$ | {20, 30 ,40} | 30 | |
| weight $\rho$ in PPO-STS | – | | {0.97, 0.98, 0.99} |
| Imaginary horizon $T_p$ in PPO-MBS | – | {30, 50 , 80} | – |

Table 1: Hyperparameters in the training process

## A.3 DETAILS OF MODEL-BASED LEARNING IN PPO-MBS

### A.3.1 DYNAMIC MODEL LEARNING

We represent the dynamic model $f_\phi(a, s)$ with a 2-hidden-layer MLP with hidden sizes $512 \times 512$ and `ReLU` as the activation function. We train the model with Adam (Kingma & Ba, 2014) optimizer with learning rate 0.0003 using a batch size of 512 for every 5 policy optimizing iterations. The model is trained over a subset randomly drawn from the training dataset with size 25,000 for 10 passes and we use validation dataset for early stopping (We evaluate the validation loss for each model learning pass). Empirically we find resetting the model parameter $\phi$ every 15 policy optimizing iterations helps improve to the performance.

### A.3.2 DATA PROCESSING

We obtain a replay buffer $\mathcal{D}$ which stores the input and the output pair $\{(s_t^i, a_t^i), s_{t+1}^k - s_t^k\}_{k=1}^N$ of the dynamic $f_\phi(a_t, s_t)$ with most recent $N = 50,000$ samples which has been used by policy learning. Further, we subtract the mean of the data and divide by the standard deviation of the data to ensure the loss function weights the different parts of the states. Finally, we split the collected data using a 3-to-1 ratio for training and validation datasets.

### A.3.3 IMAGINARY TRAJECTORY PLANNING

We use the dynamic model $f_\phi(s_t, a_t)$ and the policy $\pi_\theta(a|s)$ to collect the imaginary trajectories with a max horizon $T_p \in \{30, 50, 80\}$. To estimate $g_{\text{mb}}$, we sample a total batch size of 25,000 imaginary trajectories. To make the imaginary trajectories diverse, we randomly use the first $T_{\text{max}} - T_p$ time step of states in the real trajectory samples as initialization $s_0$ for planning, where $T_{\text{max}}$ is the maximum horizon of the environment. For *Reacher*, we only use the initial states from real trajectories $s_0$ as the initialization since its maximum horizon is $T_{\text{max}} = 50$.

### A.4 DETAILS OF ENVIRONMENTS

Here we list the reward function used in model-based dynamics learning. The environments we use are based on the top of OpenAI Gym(Brockman et al., 2016). We modified the environment by adding the observations of Mujoco(Todorov et al., 2012) environments which are expected by gym but are requisite for reward calculation.

The reward functions $r(s_t, a_t, s_{t+1})$ and the optimization horizon $T_{\text{max}}$ are described below:

| Environments | $r(s_t, a_t, s_{t+1})$ | $T_{\text{max}}$ |
|:---:|:---:|:---:|
| *Swimmer* | $s_t^{\text{xvel}} - 0.0001\|\|a_t\|\|_2^2$ | 1000 |
| *Walker2d* | $s_t^{\text{xvel}} - 0.001\|\|a_t\|\|_2^2 + 1$ | 1000 |
| *Hopper* | $s_t^{\text{xvel}} - 0.001\|\|a_t\|\|_2^2 + 1$ | 1000 |
| *Pusher* | $-0.5\|\|x^{\text{body}} - x^{\text{arm}}\|\|_2^2 + \|\|x^{\text{body}} - x^{\text{goal}}\|\|_2^2 - 0.1\|\|a_t\|\|_2^2$ | 100 |
| *Reacher* | $-\|\|x^{\text{finger}} - x^{\text{goal}}\|\|_2^2 - \|\|a_t\|\|_2^2$ | 50 |

Table 2: Detail of the environments used in our experiments

$s_t^{\text{xvel}}$ denotes the $x$-axis velocity at time $t$, which is calculated by $s_t^{\text{xvel}} = \frac{s_{t+1}^{\text{xpos}} - s_t^{\text{xpos}}}{dt}$, where $dt = 0.02$ in the Mujoco (Todorov et al., 2012) simulator. $x^{\text{body}}$, $x^{\text{arm}}$, $x^{\text{finger}}$ denote the position of the body, the arm and the fingertip of the object separately, and $x^{\text{goal}}$ denotes the position of the goal.

## B PPO-STS ON HUMANOID AND ANT ENVIRONMENTS

We compare PPO-STS with PPO on higher dimensional tasks like Humanoid and Ant, and show the results in Figure 4. It shows that PPO-STS can be consistently better than PPO, which makes PPO-STS more valuable since it does not introduce additional computational costs.

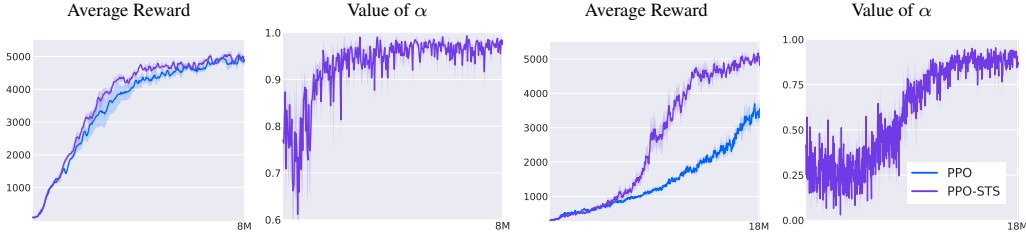

Figure 4: Comparison of PPO-STS with PPO on Humanoid and Ant.

