# OpenReview forum: "Shrinkage-based Bias-Variance Trade-off for Deep Reinforcement Learning"
_ICLR.cc/2019/Conference_

### Official Review · AnonReviewer3 · 2018-11-02
**Well motivated use of James-Stein estimator to RL problems**

**Rating:** 5
**Confidence:** 3

**Review:**

The paper suggest a shrinkage-based estimator (James-Stein estimator) to compute policy gradients in reinforcement learning to reduce the variance by trading some bias. Two versions are suggested: The on-policy gradients is shrinked either towards (i) model based gradient, or towards (ii) a delayed average of previous on-policy gradients. Empirically, both methods have better performance than the baseline.

The paper is clearly written and well motivated. Some details are lacking that would be of interest to the reader and to make the results reproducible. For example how is \hat Q estimated? The trick that is referred to in the end of page about only simulating short horizon trajectories deserves more detail. I would suggest providing more details, in the text and/or in the two algorithms.

The authors claim that JS estimator for gradient estimation in RL has not been used before. I am also not aware of any other work, but have also not been looking after that line of work. The paper seems to be a good contribution to the ever increasing literature of how to improve deep RL.

Minors:
\hat \theta on RHS in eq (7) should be \bar \theta ? Otherwise, what is \hat \theta?
section 4.2 Ww -> We

======= After revision =========

I still think this is a very interesting, novel and relevant idea that desires attention. However, on the same time, I agree with the points raised by the other two reviewers which are all well-motivated and relevant concerns. Therefore, I join the view that the paper is not yet ready for publication but I do encourage the authors to improve their work and resubmit to another venue.

---

> ### Author Response · Authors · 2018-11-27
> **Response to Reviewer3**
>
> We would like to thank the reviewer's valuable comments.
> We believe our work is the first one to utilize shrinkage estimators to improve deep RL algorithms. We will add more solid empirical experiments and more discussion about both the details and the theoretical properties in the RL settings in the next version.

---

### Official Review · AnonReviewer1 · 2018-11-03
**Promising, yet inconclusive evaluations**

**Rating:** 4
**Confidence:** 4

**Review:**

This paper presents two algorithms that improve on PPO by using James-Stein (JS) shrinkage estimator. The first algorithm, PPO-MBS, combines low bias of on-policy methods with low variance of model-based RL algorithms. The second, PPO-STS, uses JS to create a statistical momentum and reduce variance of the PPO algorithm. Both algorithms are evaluated on Mujoco environments aiming to show improvements in average cumulative reward, reduced bias and variance.

The paper’s topic is highly relevant, as the authors point out, the current state of the art in RL is largely divided between model-based and model-free methods. This paper aims at bridging the gap, and taking advantage of both sides. The writing is clear and concise, with all the math properly introduced. The proposed methods are interesting and novel. As far as I am aware, this is solid and unexplored approach with potentially significant impact.

There are two concerns with this paper. First, the evaluation results are promising, yet not fully convincing. It appears that 5 million steps is not sufficient for the policy convergence for any of the tasks (average reward keeps increasing). At the same time the variance gap (Figure 1) is reducing.
- As the training continues, does the variance for PPO-MBS become larger?
- Similar question for the average reward - the advantage of the PPO-MBS vs. comparison methods seems to be reducing. What happens with the additional training?
- How do the trajectories and behavior of the Walker2D (and other) look with PPO-MBS vs. others - is the higher average reward indicative of the qualitatively better behavior?
- Why do Walker2D and Hopper increase \alpha over time, while Swimmer and Reacher lean more towards mode model based policy over time?
- Is there something significant about the structure of the problems?
- Similar questions arise from the evaluation of the PPO-STS - it appears that the training is even less complete in this case, rendering the conclusions about the quality of the learned policy at convergence invalid.
- Why are Humanoid and Ant not evaluated on PPO-MBS? The authors should extend the training to convergence on all problems, and present the results including movies of example trajectories of all environments for both algorithms in the supplementary material.

Second, the paper introduces two competing methods.  While, the authors compare them directly,  they do not discuss how the two methods relate to each other. In the Reacher and Hopper task seems like all three methods perform about the same, and in the Hopper PPO-STS even performs worse than the baseline. This makes it difficult to assess the significance of the exposition. When should one be used, and when the other one is better?

The authors should consider either a) splitting the paper into two focusing each paper on a single algorithm with more in-depth evaluations and discussions, or b) combining the two algorithms into ones. In any case, further analysis that illuminates when the methods should be used, and how they improve the training are needed.

---

> ### Author Response · Authors · 2018-11-27
> **Reponses to AnonReviewer1**
>
> Thank you for the valuable review and suggestions for improving the paper. Followings are the detail responses to the questions.
>
> *--Is 450k steps enough for Mujoco Tasks
> We only evaluate the Reacher with 450k steps since the environment is very simple and 450k steps are sufficient to learn a good policy.  We use  4500k or more samples for all the other tasks according to their difficulty.
> We find the current steps is enough to demonstrate the advantages of our method, but will further add steps to further confirm the results.
>
> *--Performance for PPO-MBS
> In Figure 1, we demonstrated that our proposed method learns faster and can reduce variance during the training process. It is natural that the variance gap between our proposed method and baseline becomes smaller during the training since $\alpha$ increases on walker2d and the combined gradient estimator is more similar to the unbiased gradient.  As for PPO-MBS vs constant $\alpha$, the adaptive $\alpha$ is close to $\alpha=0.5$ during the training process. This also shows that we can adaptively get better performance than fixed coefficient without hyper-tuning.
>
>
> *--Why does $\alpha$ increase on Walker2d and Hopper, but decrease on  Swimmer and Reacher?
>
> This might be because that the trajectory lengths of Walker2d and Hopper increase during the training process, which increases the sample complexity gradually; this makes it difficult for the dynamic model to learn the real transition process and hence yields an increasing $\alpha$.  This may also indicate that our adaptive strategy can check the performance of the learned dynamic model automatically.
>
> *--Comparing and Combining PPO-MBS and PPO-STS
> We will add results on comparing and combining  PPO-MBS and PPO-STS in the revised version. The combination of these two approaches can be done straightforwardly with multiple shrinkage methods, which we will also investigate.
>
> At last we would like to thank the reviewer for pointing our potential issues and we will improve them in our next version.

---

### Official Review · AnonReviewer2 · 2018-11-05
**Direct Application of a Well Established Statistical Method without Theoretical Gurantees and Very Limited Emprical Support**

**Rating:** 4
**Confidence:** 2

**Review:**

The paper claims that a combination of policy gradients calculate by different RL algorithms would provide better objective values. Main focus of the work is to devise and adaptive combination scheme for policy gradient estimators. Authors claim that by using the statistical shrinkage estimators combining different gradients that have different bias-variance trade-off would provide better mean-square error than each of those individual gradient estimators. The key observations made by the authors are that gradients computed by on-policy methods would provide nearly unbiased estimators with very high variance while the gradients obtained by the off-policy methods in particular model based approaches would provide highly biased estimators with low variance. Proposed statistical tool to combine gradients is James-Steim shrinkage estimator. JS estimator provides strong theoretical guarantees for Gaussian cases but some practical  heuristics tor more complex non-Gaussian cases. Authors do not discuss  whether the JS estimator actually suitable for this task given the fact that strong assumptions of the underlying statistical approach is violated. They also do not go into any discussion about theoretical guarantees nor they provide any exposures or intuitions about that. The scope of the experiments is very limited. Given the fact that there is no theory behind the claims and the lack of strong evidence I believe this paper does not cut the requirements for publication.

To improve please add significantly more empirical evidence, provide more discussion about theoretical ground work and discussion about the suitability of the JS estimators when its required assumptions are not satisfied.

---

> ### Author Response · Authors · 2018-11-27
> **Reponses to AnonReviewer2**
>
> We would like to thank reviewer's valuable comments. As for theoretical gurantees,  JS estimator was originally motivated for normal distributions for which nice inequalities can be established to guarantee the decrease of MSE.  However, it is easily applicable to more general cases and provides a simple yet powerful strategy for addressing the {very challenge of bias-variance trade-off} that is crucial in many components of RL. The goal of this work is to fill this gap between RL and JS literature. Note that the decrease of MSE in our experiments is already a direct evidence of the effectiveness of JS in RL.  In the future we will try to use recent extended efficient shrinkage estimators in parametric models (hansen, 2016), which relaxes the normal distribution to general asymptotic distributions.
>
> Hansen, Bruce E. Efficient shrinkage in parametric models.Journal of Econometrics, 190(1):115–132, 2016.

---

### Meta-Review · Area_Chair1 · 2018-12-14
**Nice work with potential, but contributions need to be strengthened**

**Confidence:** 4
**Recommendation:** Reject

**Metareview:**

The paper introduces the use of J-S shrinkage estimator in policy optimization, which is new and promising.  The results also show the potential.  That said, reviewers are not fully convinced that in its current stage the paper is ready for publication.  The approach taken here is essentially a combination existing techniques.  While it is useful, more work is probably needed to strengthen the contribution.  A few directions have been suggested by reviewers, including theoretical guarantees and stronger empirical support.